# Grandparental childcare and second births in China

**Jing Zhang**, **Tom Emery** *

Department of Public Administration & Sociology, Erasmus University Rotterdam, Rotterdam, The Netherlands

* tom@odissei-data.nl

**Data Availability Statement:** The data is available for download from Peking University: https://opendata.pku.edu.cn/dataset.xhtml?persistentId=doi:10.18170/DVN/45LCSO

**Funding:** The authors received no specific funding for this work.

## Abstract

China has low birth rates at higher parities and intensive grandparental childcare. Despite this, there has been little empirical research into the role of intergenerational support in the transition to second birth. This study examines whether grandparental childcare increases the likelihood and speed of a transition to second birth in the context of relaxations in Chinese family planning policy and whether this differs for working and non-working mothers. Using data from the China Family Panel Studies (2010–2016), the association between grandparental childcare, mother's working status and second childbirth are explored using split-population survival models to distinguish between the impact on the timing of fertility and ultimate parity progression. The odds of having a second child are four times higher for those who use grandparental childcare than those that don't. Amongst those who have a second child, grandparental childcare leads to 30% lower odds of transition to second birth than those without grandparental care, each month. Grandparental childcare is also associated with maternal employment, which is itself associated with a sharp decrease in the transition to second birth. At the micro-level, grandparental childcare helps mothers continue working, which in turn defers a second birth. The results emphasise the importance of work-life balance strategies, such as grandparental care, in enabling women of childbearing age to realize their fertility intentions in combination with work.

## Introduction

Recent relaxations in Chinese family planning policies aim to increase the rate of higher order births. The extent to which these efforts are successful will have a substantial influence on Chinese and global demographic projections, and the environmental and economic scenarios these projections are associated with. Given this, there has been surprisingly little attention paid to determinants of transition to higher order births in China. There is a particular lack of evidence on the role of grandparental childcare in transitions to higher order births given the substantial role that grandparents play as childcare providers in China. This analysis investigates whether there is an association between grandparental childcare and the transition to a second child, and the extent to which this differs depending on the mother's labour force participation.

**Competing interests:** The authors have declared that no competing interests exist.

The total fertility rate in China has been around 1.6 or lower since 1997 [1, 2]. Even though the one-child policy was in place until 2015, large parts of the rural and non-han population have enjoyed exemptions to this policy for a long time. If both father and mother come from a single-child family, they are also allowed to have a further child and higher order births have been increasing. According to the 2017 China Fertility Survey, there was a 23% decrease in first order births and a 65% increase in the second order births from 2011 to 2016 [2]. Since 1990, Chinese female labour force participation rates have remained high [3], and public-funded childcare programs have been curtailed through socio-economic reforms [4–6]. Due to inadequate childcare services, absence of parental-leave policies and non-flexible working schedules, grandparents step in to look after their grandchildren and/or do household chores [7, 8]. Chinese grandparents are more likely to live with their children than in Europe or North America, and 50.4% of Chinese children under sixteen years old lived in three-generation households in 2010 [9, 10]. More than half of people aged 55 or over provide care to their grandchildren, and many of them are actively involved and assuming the primary caregiving role whilst parents are at work [11–13].

A growing body of research has shown that women living with parents or in-laws are more likely to *plan* a second-child [14–16]. However, these differences in fertility plans have been small or not statistically significant [17], particularly in low fertility areas [18]. Findings from studies specifically examining grandparental childcare rather than just co-residence are more consistent, showing that the availability of childcare support from parents and in-laws increases the likelihood that a woman *intends* to have more children [18, 19]. To date, no study has examined the role of grandparental childcare in the *realisation* of a second birth in China using nationally representative data. Wang & Zhao (2022) examined the role of grand-parental care in the transition to second births amongst internal migrants. They found that grandparental childcare increased the likelihood of transitioning to second birth, even when accounting for the increased likelihood of maternal employment that grandparental childcare provides. Using nationally representative data, this paper corroborates and extends this analysis to a representative sample of the entire Chinese population. Furthermore, this analysis considers the impact of the timing of employment and the transition to second birth to better understand the interdependency between employment and fertility.

The paper focuses on second births for two reasons. First, higher-order births are of growing scientific and policy interest in the Chinese context due to the changes in Chinese family planning policy allowing couples to have two or more children. It has been argued that the low fertility rate in China is not only due to family planning policies such as the one-child policy but also due to broader socio-economic developments, which make the opportunity costs of children prohibitively high [20]. Given this, it is unclear to what degree changes in family planning laws will change birth rates, and a better understanding of determinants of parity progression in China is needed. There is extensive literature on the transition to parenthood in China but little on higher-order births [21–24]. This study is the first to use nationally representative data to examine the realised second-births instead of second-birth intentions or desires. This distinction is vital as few Chinese women realise their desired family size [25].

Second, studying the transition to higher parity focuses on existing childcare support practices and their role in shaping subsequent fertility behaviour, as opposed to *hypothetical* and *planned* childcare arrangements ahead of a transition to the first birth. With higher-order births, parents better understand the realities of grandparental childcare. New parents are very aware of the complex interplay between various generations and the extent to which intergenerational support capacity is time-bound by multiple actors in a way that expecting parents are not [26]. The timing of a second child is potentially influenced by whether it aligns with the working life of the mother, the growth of the first child, and the working life and possible health concerns of

the grandparents who are potential childcare providers [7, 15, 16, 27]. Given the interdependency between generations that is apparent through the existing childcare arrangements of the first child, this study is guided by the life course approach, which emphasises the multidimensionality of the linked life domains and time dependency of the life course [28].

The empirical analyses use longitudinal data from China and a split-population event history model with time-varying covariates to identify the influence of grandparental childcare on the transition to a second child and its timing. The issue of grandparental childcare is particularly crucial in China because the conflict between family and work for women maybe even more challenging with the relaxation of family planning policies that have occurred throughout the last decade. We therefore examine the extent to which grandparental support operates differently for working and non-working women. The low fertility and unrealised fertility aspirations in China require a broader understanding of the potential determinants of fertility. The findings of this paper provide an insight into how grandparental childcare relates to these on-going challenges.

## Theoretical framework and hypotheses

This section sets out the following key concepts: linked lives, interdependent life-domains and the time dependence of the life course. The linked lives principle embeds the personal life course in the extended family and a broader social context [28, 29]. Interdependent life-domains refer to the association between outcomes in various areas of life, the most important in this study being family and work. Time dependence refers to the interdependence of individuals' experiences of past, present and future and the ways in which these interact in the inter-birth intervals. The present study focuses on grandparental childcare as a social relationship with simultaneous consideration of the time dependency and overlap of family and work domains in the specific Chinese cultural and family context.

### Linked lives

Parents and in-laws are linked to a woman's fertility through the transmission of genes and social norms, the communication of expectations and family values, and through the household structure or alloparenting behaviours as a source of social capital in the form of support [30]. So far, empirical studies on the relationship between grandparents and female fertility were primarily conducted in western countries [31–37], East Asia [15, 38–41], and some developing countries in South Asia [42, 43]. These findings emphasise the role of extended family in fertility, suggesting that, overall, grandparents availability was correlated with higher fertility [44]. However, the availability of grandparents was typically measured by survival status or co-residence with women. Most of the empirical research on East Asian populations typically measured the impact of co-residence with parents' in-law on women's fertility. The contribution that specific elements and acts of grandparenting make to women's fertility is still unclear. When direct grandparental childcare has been considered, the evidence from western countries showed a mixed pattern [44]. Only one study on Asian countries by Snopkowski and Sear [42] measured direct grandparental help in the Indonesian population. They found that grandparental help did mediate the positive associations between the presence of a grandparent generation and fertility, and the relationship was the strongest when the help came from the woman's mother-in-law.

Grandparental childcare is a complex process in which the availability of potential caregivers, the needs of the mother and the child, and cultural norms and expectations all play a crucial role [45]. Chinese family norms, like other Asian societies, attach substantial importance

to paternal lineage [46]. Paternal grandparents are more likely to provide grandchild care and other services to their adult sons [47, 48].

Historically the parent-child relationship in China follows a Confucian tradition in which family continuity is one of the primary principles [12]. Fertility preferences and the desired number of children are driven by these specific family values and expectations [18]. Grandparents and in-laws are engaged in daily communications of expectations for more offspring. Thus, the fertility decision is not only a dyadic decision between couples but also influenced by the extended family [14]. Meanwhile, grandparents and in-laws share childcare duties and make economic and non-economic contributions to the family [49–51]. Following this reasoning, one would expect that Chinese mothers receiving grandparental childcare would be more likely to have a second child than those who are not (Hypothesis 1). It could, however, be contended that receiving grandparental childcare may indicate resource requirements from a wider extended family and therefore, potentially, invite a 'veto' by that extended family on further children. This would suggest that grandparental childcare be associated with a lower transition to second birth.

## Time dependence of the life course

The fertility literature focuses on the timing and the spacing of parities [29]. The cooperative breeding hypothesis indicates that alloparental care, like grandparental childcare, may lead to shorter birth intervals for humans than other mammalian species [52]. However, most of the empirical evidence is from traditional agricultural societies [44], and no associations for the second birth have been found [33]. A few empirical studies on Asian populations have found a consistent influence of co-residence with parents or in-laws on shorter interbirth intervals [15, 38–40]. With a direct measure of grandparental help, the question in this study is whether the experience of grandparental childcare for the first child could accelerate or delay additional childbearing.

In line with the cooperative breeding hypothesis, the influence of grandparental childcare should be similar to that of grandparental co-residence mentioned above. Furthermore, with the increasing age at marriage and entry into parenthood in China [53], the fertile period is limited for women. This may lead to women who would like to have more children "racing against the biological clock" [54]. Given this, we expect to find that women who are receiving grandparental childcare help for their firstborn child will experience an earlier second birth (Hypothesis 2). Again, as with the first hypothesis, if grandparental childcare would be considered as signifying an auxiliary role that indicated resource constraints in the couple as in many western contexts, it could be expected that grandparental childcare be associated with a slower transition to a second child.

## Interdependent life-domains

Childbearing over the life course also intersects with experiences and decisions in other life domains [55]. Research consistently indicates that female labour force participation delays entering parenthood, increases birth intervals, and is negatively associated with the eventual number of children [56, 57]. Reasons for working mothers being less inclined towards additional births include the additional cost of raising existing children [58], the opportunity costs [59] and personal work-family preferences [60]. With regards to the labour market in China, regulations of workplace and wage policies increasingly demand full-time work and even longer working hours [19, 61]. As a consequence, mothers who need childcare are forced to choose between not working or full-time work with limited leave [4]. This suggests that mothers may decide to postpone having a second child or even consider abandoning additional

births if they are currently working (Hypothesis 3). Conversely, maternal employment will improve the income of the household and potentially overall extended family and thus increase the ability to afford further children.

This hypothesis should be considered in relation to grandparental childcare support. Many studies have found a positive relationship between grandparental childcare support and changes in mother's labour force participation [12, 32, 62–65]. Recent research has found that the influence of grandparental childcare on sustaining maternal employment is even greater than that of formal childcare services in urban China [27]. Therefore, grandparental childcare could positively affect maternal employment and subsequently postpone or even negate further additional births, in contradiction to hypotheses 1 & 2.

Women's aims and ambitions with regards to both work and family are interrelated in their contribution to individual well-being [66]. Considering women's intentions and motivations [60], mothers might differ in the way that they perceive grandparental childcare help as means for work-life balance. On the one hand, grandparental childcare driven by maternal employment may prevent the opportunity to have a second child, which is referred to as a compensation effect [28]. Here, maternal employment may act as a suppressor/mediator for the relationship between grandparental childcare support and fertility. Thus, one would find the negative effect of maternal employment offset by the positive effect of grandparental childcare on the likelihood and timing of the second birth (Hypothesis 4a).

**Table 1. Descriptive statistics for variables in the target population, and analytical sample.**

|  | Whole Sample | Analytic sample |
|---|---|---|
| *Number of individuals* | *N = 5,045* | *N = 2,097* |
| Second birth occurred after first interview (%) | 17.1 | 25.8 |
| **Women's characteristics** |  |  |
| Age in months (S.D.) | 348.8 (63.5) | 390.9 (64.2) |
| Only child (%) | 11.0 | 10.1 |
| Educational level (%) |  |  |
| *Primary school or below* | 25.7 | 29.1 |
| *Lower secondary school* | 37.9 | 38.9 |
| *Higher secondary school* | 18.4 | 16.9 |
| *Tertiary education/above* | 18.0 | 15.1 |
| Husband's educational level (%) |  |  |
| *Primary school or below* | 19.7 | 22.4 |
| *Lower secondary school* | 41.1 | 42.5 |
| *Higher secondary school* | 19.9 | 18.5 |
| *Tertiary education/above* | 19.4 | 16.6 |
| **Household characteristics** |  |  |
| Rural hukou (%) | 68.8 | 65.2 |
| Log household income per capita (S.D.) | 9.2 (0.9) | 9.2 (0.8) |
| **First child characteristics** |  |  |
| Boy (%) | 54.1 | 56.7 |
| **Key independent variables at first interview** |  |  |
| Working (%) | 53.0 | 66.3 |
| Grandparental childcare (%) | 27.8 | 30.9 |

Source: CFPS 2010–2016.

On the other hand, the positive effect of grandparental care on maternal labour force participation may allow mothers to reconcile conflicting work and family ambitions, which is referred to as the spill-over effect [28]. This suggests that a working mother with grandparental childcare would be more likely to have a second child and to shorten the space between births than working mothers without such support, as grandparental childcare enables mothers to manage family and work together (Hypothesis 4b).

## Data & methods

### Data source and sample construction

The data used for this study are from four waves of the on-going national representative longitudinal survey programme the China Family Panel Studies (CFPS). The first wave was conducted in 2010 with an original sample of 14,960 households in 25 provinces and regions, representing 95% of the Chinese population [67]. The respondents were followed bi-annually (2012, 2014 and 2016) to track changes in the characteristics of households' and individuals' social and economic activities and attitudes. All household members were interviewed, and information about family relations, individual's working status, and women's fertility history were collected. One advantage of this data set is that CFPS collected information on the primary childcare provider for children younger than 16 in each round. We were, therefore, able to examine whether grandparental childcare influenced subsequent childbearing. It should be noted that this is not the same as western studies of grandparenting, where childcare provision is auxiliary to the role of parents and formal childcare. In the CFPS, grandparenting is being measured as the *primary* childcare provider.

To examine the relationship between grandparental childcare and second-births, we selected a target sample of married women with only one child and under the age of 40 when entering the survey for the first time. We removed women whose child passed away (N = 22) or who were not in their first marriage (N = 401) because the childbearing decisions and grandparental childcare opportunities would likely differ in women who experienced such life events. Both events are rare in China. Twin births were also not considered in the analyses. Dropping respondents with missing values in key variables results in a sample of 2,097 women. Table 1 presents the descriptive characteristics of the whole sample population and the analytical sample.

### Dependent variables

The dependent variables of interest are the likelihood and timing of the second birth. The processing time for the second birth was measured in months since the birth of the first child. If there was no second birth, the observation ended either at the last interview, when the mother reached 40 years old or after the child reaches 16 years old because both the need for childcare and the probability of giving additional birth is rare after such a prolonged duration from the first birth. Of the 2,097 one-child mothers who entered the observation window, 542 (27%) proceeded to have a second child and the mean of time at risk is 44.28 months (3.69 years).

### Independent variables

Grandparental childcare is a dummy variable that indicates whether mothers received childcare help from parents or the in-laws. This study focusses on intensive grandparental childcare where grandparents play the primary caregiver role. Information on whether the child was primarily cared for by the grandparents was collected from the children's questionnaire in each wave and matched to their mothers. Women with children who are mainly cared for by the

grandparent were categorised as 'receiving grandparental childcare support', whereas those women with children not cared for by the grandparents were categorised as 'no grandparental childcare support'. Relative to the existing literature, this therefore reflects an intensive level of grandparental childcare support. Such intensive childcare is common in China with 30.9% of the analytical sample reporting such support.

Mother's working status was constructed from the work-related questions in the individual questionnaire, including both the main job and secondary jobs. The constructed categorical measure of employment was generated by the CFPS team [67]. In this study, women not in the labour force or unemployed were grouped and categorised as not working, whereas women who were currently employed (including those on maternal or short leaves) were categorised as working (66.3%). The percentage of mothers using grandparental childcare was higher for working mothers (35.5%) than for those not working (21.8%) (Not shown in table).

CFPS, like many other longitudinal social surveys, asks respondents about their childcare experiences and working status between two interviewed waves. Together with information on the timing of second birth, we restructured the two time-varying independent variables so that there is a record for each episode (in months).

## Control variables

The models contain several control variables that studies have shown to influence the second birth. Age is a strong predictor with fertility. We included mother's age in months as a time-varying variable in the models. The socio-economic measures relied on three time-fixed variables: education of women and their husbands, and household income. Women's educational attainment was categorised as primary school or below, lower secondary school, higher secondary school, and tertiary education or above. The husband's highest educational attainment was also included. Household income was calculated as the mean of the interviewed waves and was log-transformed in the analytical models. China's family planning policies and social welfare system vary according to the household registration type (hukou), so a dummy variable measuring rural or urban hukou at the first interview was included in the analyses. Whether the mother is the only child was also included in the model, as the preference for a particular family size may transfer across generations [19, 64] and may also be associated with grandparental support. Regarding the existing child's characteristics, the gender of the first child was included as a crucial indicator of additional childbearing as there is a strong son preference in China [25].

## Analytic strategy

The present study applied event history analysis using parametric split-population survival models to investigate how grandparental childcare support was related to the likelihood and timing of the second birth. Traditional conventional survival models such as Cox proportional regression assume that everyone would eventually experience the event of interest if follow-up is long enough, or are counted as censored [68]. However, this is certainly not the case with second births, especially in the context of China, where family planning policies regulate women's fertility behaviours and the total fertility rate is far below replacement [1]. Furthermore, the proportion of women who never have a second child would have distinct characteristics compared to those who eventually have additional children and prevent estimation of the speed at which a transition is made. We therefore analysed these segments of the population independently.

With the assumption that a proportion of the population will never experience the event of interest, the split-population models consist of two parts, which jointly estimate a parametric

model for a fraction of the population that never experience the second birth and a parametric survival model for the timing of the second birth conditional on a second birth occuring [68–70]. STATA 16 and the user-written add-on module CUREREGR [71] were used for estimating the models. The likelihood fraction is modelled using a logistic link. Regarding the parametric survival model, two types of split population models, mixture and non-mixture models, and different distributions of the time patterns of the second birth were considered [70]. The survival model makes two basic assumptions: 1) that the hazards across strata are proportional and 2) that the relationship between the log hazard and independent variables is linear. The goodness-of-fit indicators AIC and BIC were used to choose the most appropriate model. Among the models estimated, the mixture model combined with the lognormal distribution family for the time patterns consistently provides the best fit to the data; therefore, it was chosen as the final model and the results estimated from this type of model were reported in the next section.

To test the research hypotheses, a set of mixture lognormal models are constructed. First, we examine how grandparental childcare is associated with the likelihood of having a second child and the waiting time until the second birth (Model 1). Next, Model 2 estimates how mother's working status is related to the likelihood and timing of the second birth. Then, Model 3 includes both grandparental childcare and mother's working status examining whether the associations of grandparenting change once the employment status is considered. An additional logistic model with mother's working status regressed on grandparental childcare and covariates is estimated to identify the association between grandparental childcare and mother's working status. Finally, we turn to differences in the grandparental care and fertility by mother's working status by including an interaction term (Model 4).

## Results

### Descriptive statistics and changing patterns of having a second child

Fig 1 represents the time to the second birth through non-parametric Kaplan-Meier analyses by grandparental childcare and mother's working status. The survival estimates show that mothers with grandparental childcare are more likely to experience a second birth sooner than mothers without such support. For mothers who had a grandparent as the primary caregiver, it is estimated that it would take 40 months for 25% to have had a second birth, compared to 48 months for mothers without such support. Working mothers were far less likely to have a second child, with it taking an estimated 58 months for 25% of mothers to have had a second child. For mothers who were not working, this was just 32 months. The shape of these four curves is also notable, with non-working mothers exhibiting a much sharper gradient representing much shorter intervals between births.

The estimated hazard functions show the risk of second birth increases and reaches a peak around 48 months, then decreased as time passes (Fig 2). This pattern also holds for different grandparental childcare situations and mother's working status. The right-skewed inverted U-shape hazard distribution supports the lognormal model chosen by this study, which commonly features an increase followed by a decrease in the hazard function over time [68, 70].

### Grandparental childcare, mother's work and second childbirth

This section presents the relationship between grandparental childcare, mother's working status and second birth progression from the split-population models. Table 2 shows the estimated parameter coefficients for each model described in the method section. Full results can be found in the annex (Tables 3 and 4). The likelihood component in each model estimates the predictor effects on not having a second birth. Somewhat counterintuitively, a negative

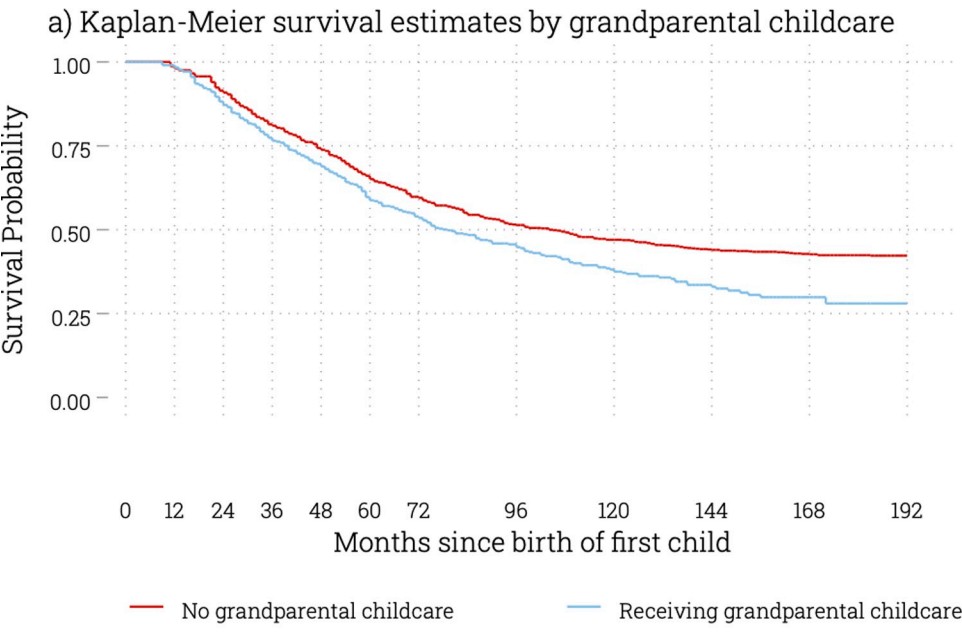

a) Kaplan-Meier survival estimates by grandparental childcare

No grandparental childcare — Receiving grandparental childcare

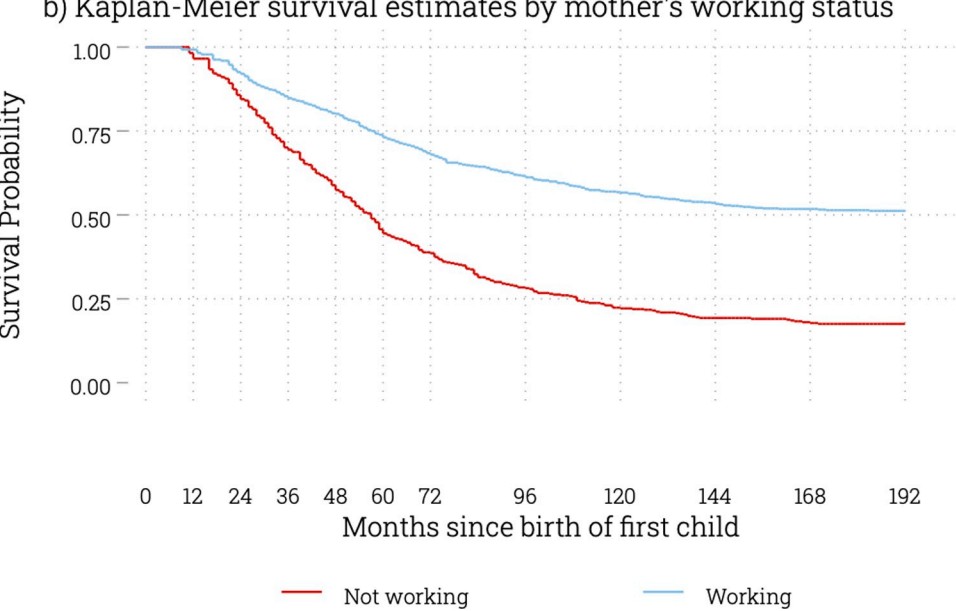

b) Kaplan-Meier survival estimates by mother's working status

Not working — Working

**Fig 1. Kaplan-Meier Survival Curves for the transition to second birth.**

coefficient indicates that women are at higher risk of a second birth as the model estimates the likelihood of not having a second child during the observation period. The duration results provide the estimates for the speed-of-progression parameters in each model. A positive coefficient in the lognormal duration model indicated a slower progression to the second birth, given that a woman had a second child. This is because a positive coefficient indicates an increased likelihood of not having a childhood in a specific time period, given that they do have a second child over the observation period.

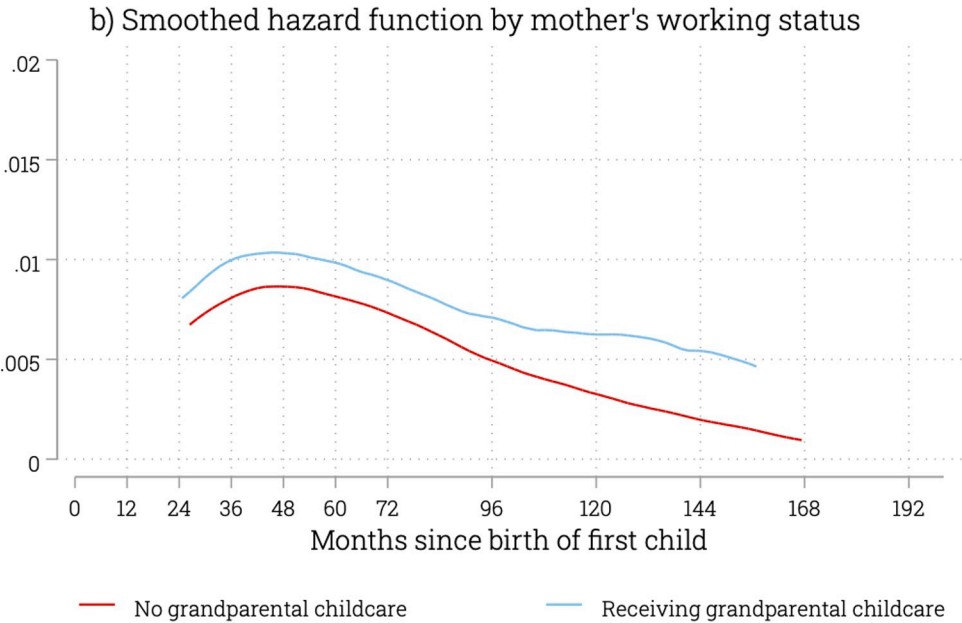

b) Smoothed hazard function by mother's working status

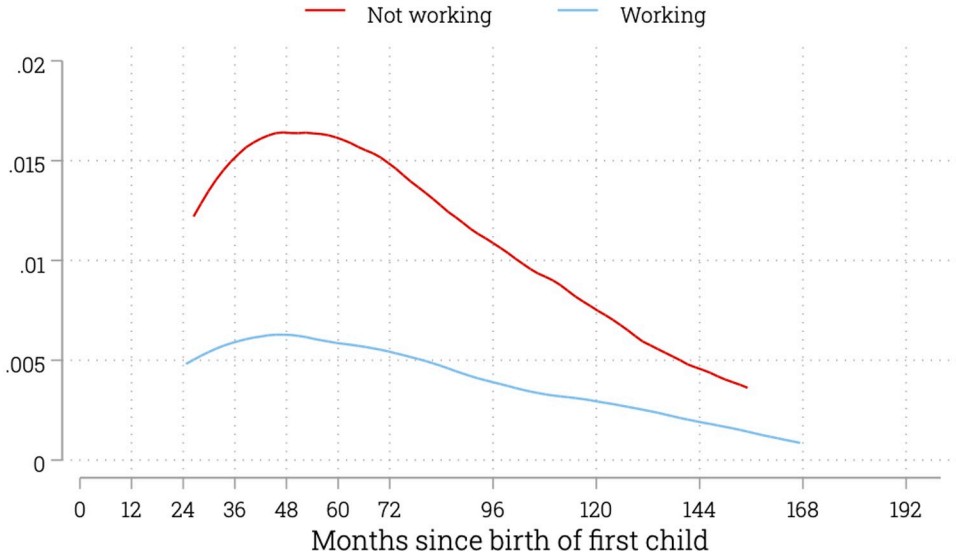

**Fig 2. Hazard function for the birth of a second child.**

Model 1 estimates the association between grandparental childcare and the transition to a second birth without including the mother's working status. The results from the likelihood component show that women receiving grandparental childcare help for their first child have a higher probability of progressing to a second birth than mothers without grandparental childcare help (OR = 0.24, $p < .01$). This indicates that the odds of having a second child are four times higher for those who use grandparental childcare than those that don't. For those who progressed to a second birth however, there is no evidence that grandparental childcare help is associated with how quickly the second birth occurred.

**Table 2. Summary of the estimated coefficients from split-population models assessing progression to and timing of the second child.**

|  | Model 1 | Model 2 | Model 3 | Model 4 |
|---|---|---|---|---|
| **Likelihood of Survival (no second birth)** |  |  |  |  |
| Grandparental childcare | -1.43** |  | -1.54*** | -2.69 |
| Working |  | 1.75** | 1.88*** | 1.82*** |
| Grandparental childcare # Working |  |  |  | 1.26 |
| **Timing of Failure (the second birth)** |  |  |  |  |
| Grandparental childcare | 0.13 |  | 0.21** | 0.35** |
| Working |  | -0.36*** | -0.44*** | -0.30** |
| Grandparental childcare # Working |  |  |  | -0.26 |
| **Statistics** |  |  |  |  |
| AIC | 6165.30 | 6120.08 | 6092.39 | 6092.10 |
| BIC | 6335.16 | 6289.94 | 6274.83 | 6287.13 |

Source: CFPS 2010–2016.

*Notes: All models include covariates in Table 1.*

* $p < .05$

** $p < .01$

*** $p < .001$.

Considering the mothers' working status, Model 2 shows a strong negative association of mother's working status with having a second child (OR = 5.76, $p < .000$), without considering the grandparental childcare factor. Interestingly, the results from the duration component show that working mothers have a faster transition to second birth compared to the mothers who were not working (HR = 0.70, $p < .000$). The explanation could be that shorter birth intervals minimise interruptions in a mothers' employment career and therefore if they are going to have a child, it should be quickly [72].

Model 3 included both grandparental childcare and working status. The directions of the association between second birth and grandparental childcare and that between working status are the same as we found in Model 1 and 2. The size of the coefficients in Model 3 increased in both positive and negative terms, particularly in the duration sub-model. By adding the mother's working status in the model, the hazard ratio of grandparental childcare in the timing of second birth changes to 1.23 and became a statistically significant factor. This result indicates a slower progression to second birth for those mothers receiving grandparental childcare help compared to mothers without grandparental childcare help. Based on a comparison of the hazard ratios in different models, mother's working status suppresses 61% of the delay in the second birth in the group of mothers receiving grandparental childcare help (The suppression is calculated by the change in hazard ratios before and after including the mediator: 1- (HR adjusted/ HR unadjusted) = 1 –(0.21/0.13)). In addition, results from the logistic regression model show that grandparental childcare is significantly associated with mother's working status (OR = 2.19, $p < .000$), with mothers receiving grandparental childcare support more likely to be working.

Model 4 included the interaction terms between grandparental childcare support and mother's working status. The results showed a statistically insignificant interaction term and no improvement in the goodness-of-fit assessment. Therefore, the association between grandparental childcare help and mother's fertility is somewhat consistent regardless of the mother's working status.

**Table 3. Selection element of Split population model with all covariates (log odds).**

| | (1) | (2) | (3) | (4) |
|---|---|---|---|---|
| | Model 1 | Model 2 | Model 3 | Model 4 |
| Age | 0.0298*** | 0.0246*** | 0.0280*** | 0.0250*** |
| | (4.96) | (3.63) | (4.37) | (3.51) |
| Education | | | | |
| Lower Secondary Edu | 0.120 | 0.019 | 0.074 | -0.002 |
| | (0.31) | (0.06) | (0.18) | (-0.01) |
| Higher Secondary Edu | -0.730 | -1.054 | -1.040 | -0.998 |
| | (-1.13) | (-1.65) | (-1.57) | (-1.56) |
| Tertiary Edu | -1.604 | -1.963^*** | -2.452^*** | -2.319^*** |
| | (-1.77) | (-2.26) | (-2.54) | (-2.47) |
| Spousal Education | | | | |
| Lower Secondary Edu | -0.107 | -0.0817 | -0.0702 | -0.0510 |
| | (-0.26) | (-0.22) | (-0.17) | (-0.12) |
| Higher Secondary Edu | -0.557 | -0.401 | -0.505 | -0.438 |
| | (-0.98) | (-0.79) | (-0.89) | (-0.80) |
| Tertiary Edu | -0.629 | -0.301 | -0.622 | -0.494 |
| | (-0.78) | (-0.46) | (-0.83) | (-0.65) |
| Rural Residence | -1.738** | -1.824*** | -2.206*** | -2.067*** |
| | (-3.25) | (-3.50) | (-4.05) | (-3.45) |
| Income (Log) | 1.274*** | 1.317*** | 1.425*** | 1.468*** |
| | (4.84) | (5.39) | (4.79) | (4.82) |
| Only child | -0.464 | -0.880 | -0.500 | -0.728 |
| | (-0.52) | (-0.79) | (-0.56) | (-0.76) |
| Male | 0.552 | 0.557 | 0.493 | 0.520 |
| | (1.72) | (1.91) | (1.52) | (1.66) |
| Receiving grandparental childcare | -1.427** | | -1.537*** | -2.686 |
| | (-3.18) | | (-3.46) | (-1.26) |
| Working | | 1.751*** | 1.878*** | 1.818*** |
| | | (5.04) | (4.75) | (4.41) |
| Receiving grandparental childcare X Working | | | | 1.264 |
| | | | | (0.55) |
| Constant | -23.08*** | -22.53*** | -24.44*** | -23.58*** |
| | (-6.56) | (-6.03) | (-6.28) | (-6.01) |

## Discussion

With the relaxation of family planning policy in China, an increasing number of women are having more than one child and determinants of higher order births are a key policy concern [15]. Decisions regarding a second birth are influenced by interdependent family and work domains. Using a life course approach, this study examined the relationship between grandparental childcare support, mother's working status, and the transition to a second birth by using split-population models on a recent representative longitudinal survey from China.

First, we replicated previous studies regarding the factors associated with the transition to a second birth and addressed the unrealistic assumption that all mothers will proceed to have a second child. In line with previous research, we confirm that grandparental childcare support is associated with a higher likelihood of having a second child [15, 42]. Regarding the work domain, the traditional conflict between childbearing and work for women is also affirmed

**Table 4. Timing element of Split population model with all covariates (log odds).**

| | (1) | (2) | (3) | (4) |
|---|---|---|---|---|
| | Model 1 | Model 2 | Model 3 | Model 4 |
| Age | -0.002[*] | -0.002 | -0.001 | -0.001 |
| | (-2.00) | (-1.54) | (-1.36) | (-1.39) |
| Education | | | | |
| Lower Secondary Edu | 0.0607 | 0.0641 | 0.0640 | 0.0556 |
| | (0.66) | (0.70) | (0.70) | (0.59) |
| Higher Secondary Edu | -0.340[*] | -0.329[*] | -0.310[*] | -0.315[*] |
| | (-2.47) | (-2.38) | (-2.27) | (-2.28) |
| Tertiary Edu | -0.0896 | -0.0125 | -0.0679 | -0.0649 |
| | (-0.42) | (-0.05) | (-0.32) | (-0.30) |
| Spousal Education | | | | |
| Lower Secondary Edu | 0.00491 | -0.00914 | -0.0233 | -0.0173 |
| | (0.05) | (-0.10) | (-0.25) | (-0.18) |
| Higher Secondary Edu | -0.111 | -0.0788 | -0.0851 | -0.0834 |
| | (-0.85) | (-0.61) | (-0.66) | (-0.65) |
| Tertiary Edu | 0.162 | 0.245 | 0.211 | 0.224 |
| | (0.90) | (1.35) | (1.21) | (1.23) |
| Rural Residence | 0.341[*] | 0.332[*] | 0.314[*] | 0.303[*] |
| | (2.47) | (2.09) | (2.29) | (2.11) |
| Income (Log) | -0.279[**] | -0.204[**] | -0.251[***] | -0.222[**] |
| | (-4.10) | (-2.71) | (-3.61) | (-2.75) |
| Only child | -0.407[**] | -0.430[*] | -0.392[**] | -0.429[**] |
| | (-2.89) | (-2.53) | (-2.76) | (-2.87) |
| Male | -0.299[***] | -0.225[**] | -0.254[***] | -0.245[**] |
| | (-3.84) | (-2.81) | (-3.34) | (-3.16) |
| Receiving grandparental childcare | 0.135 | | 0.209[**] | 0.348[**] |
| | (1.69) | | (2.58) | (2.71) |
| Working | | -0.357[***] | -0.438[***] | -0.301[**] |
| | | (-3.62) | (-4.84) | (-2.62) |
| Receiving grandparental childcare X Working | | | | -0.264 |
| | | | | (-1.65) |
| Constant | -1.240 | -1.811[*] | -1.483[*] | -1.765[*] |
| | (-1.80) | (-2.48) | (-2.18) | (-2.40) |
| Observations | 3988 | 3988 | 3988 | 3988 |

*t* statistics in parentheses

[*] $p < 0.05$

[**] $p < 0.01$

[***] $p < 0.001$

(Wang & Zhao, 2022). The results indicate a negative effect of mother's working status on the likelihood of having a second child.

Moreover, we examine the timing of the second birth conditional on such a birth occurring [69], thus extending the empirical evidence on the birth interval between the first and second child. Contrary to our expectation, the analysis showed that grandparental childcare significantly *delayed* the progression to a second birth, whereas the mother's working status was associated with a faster progression to second birth.

These findings indicate a methodological problem in previous research based on classical Cox models because the validity of the interpretation is arguable when the population considered at risk includes women who will never experience the transition to a second child. The use of split-population models also identified theoretical gaps in the different mechanisms regarding both the probability of a second birth and its timing. For applied researchers and policymakers, it has significant implications for understanding the relationship between grandparental childcare and second birth in contemporary China. Using the split population model, the analysis shows that whilst grandparental care appears to accelerate transition to second birth in the population, it actually leads to a deference of transition when we look exclusively at the population that ultimately transitions. This indicates that grandparental care simultaneously increases the likelihood of transition to a second child and defers it.

The paradoxical effects of grandparental childcare suggest that having grandparental childcare support does support the transition to a second child, but also delays such births. There are several ways in which this apparent paradox could be explained. Care arrangements for children other than grandparents that are not reliant on intergenerational solidarity, such as private childcare, may be more amenable to compressed birth spacings than grandparental care. It should be noted that such delays in the transition should not be equated with low fertility desires and can instead reflect the opposite. Those with grandparental childcare support may be empowered to have greater control over their own parity progression and thus feel able to defer a birth until a time that is optimal for them. The lower overall transitions amongst those without grandparental childcare may reflect a decision structure for them in which they must choose between having a second child soon after the first or not having one at all. For policymakers looking to extend support for working families through childcare or leave policies, this must be kept in mind as such policies can lead to a decrease in birth tempo, even if the long-term quantum effect is positive.

The delays observed in the transition to second birth amongst those receiving grandparental childcare may also reflect something about the nature of the arrangements. According to Simmel [73], triadic relationships risk the potential of one member becoming subordinate to the other two thus threatening their independence and increasing ambivalence. For example, the timing of a second child when grandparental care is involved depends on the preferences of three parties rather than two, increasing the potential for delays. This would be particularly true if each party (mother, father, and grandparent) holds a veto on the decision of when to have another child. The timing must then be agreed upon by all three parties, and there is an increased likelihood of deferment. This could be particularly true if the timing decision now must also align with the life course developments and responsibilities of the grandparent. Future research and more detailed analysis on the relation between grandparental support and fertility should be explored. Particular attention should be given to the role of the grandparental life course and its interaction with the life course and timing preferences of the parent generation.

This study examined the link between interdependent work and family life domains and the influence on a mother's fertility outcomes. The associations between grandparental childcare, a mother's working status, and fertility have frequently been investigated separately. This study investigated the mediating and moderating effect of mother's working status on the relationship between grandparental childcare and the transition to a second birth. The non-significant coefficient of the interaction term between grandparental childcare and mother's working status suggests rejecting the spill-over effect of the linked life domains hypothesis [29]. Instead, the findings here lend support to the compensation effect where using grandparental childcare helps mothers continue working, which in turn defers the transition to second birth (Fig 3). However, it should be noted that the standard errors for the interaction

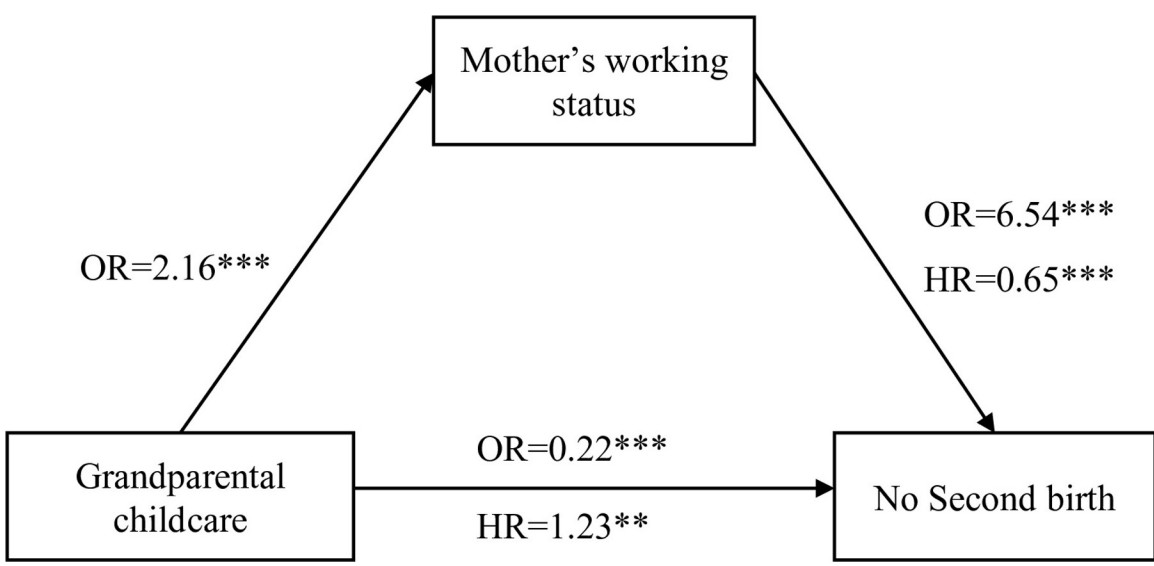

**Fig 3. Final model for grandparental childcare, mother working status, and second birth (adjusted model).** Notes: OR indicates odds ratio, and HR, hazard ratio. All results are from the fully adjusted model including covariates in Table 1.

coefficient are large and therefore caution should be taken not to over interpret this evidence in support of such a compensation effect and further research is of course required to understand better how grandparental childcare and labour force participation interact.

## Limitations

The design of this study cannot establish the direction of causality. One of the limitations of this study is that information on grandparental childcare and mother's working status was collected at the same time point in CFPS. This makes it possible that mothers' working status determines the level of grandparental childcare. Furthermore, both childcare and mothers' labour force participation are processes that unfold over time rather than as static phenomena. This also means that it remains unclear whether grandparental childcare leads to maternal employment or vice versa. In reality, these are likely two interdependent and mutually reinforcing processes. The current circumstances are built on past experiences, and experiences in different life domains are influenced by each other, which is known as 'interdependencies of parallel careers' [74]. Due to limited data, we cannot apply a counterfactual causal mediation analysis [55] with the split-population models presented in this paper. Rather than seeking a warrant for making absolute claims, the issue of causal interrelationships between these factors and the pathways of the direct and indirect influences of childcare and work on fertility over time is reserved for future work, where suitable data are more readily available.

Some other limitations of the present study should be mentioned. Only grandparents as primary childcare givers were considered in the measure of grandparental childcare support. This may limit the findings in several ways. If grandparents are reported as primary childcare givers, we assume they are highly involved and would have a crucial influence on mothers' lives. However, there could be selection bias and other unobserved heterogeneity in the types of grandparental childcare given. It is unlikely that the findings would be similar for a lower level of grandparental childcare involvement, or for using other measures such as co-residence, financial support from grandparents, contact or relationship quality between mother and grandparents. We also did not distinguish the lineage differences in this study because the sample size

for the group where maternal grandparents act as primary caregiver is too small. Thirdly, the intergenerational dynamics within China are dynamic and both population ageing, economic development, a global pandemic, and changes in the restrictions on higher-order births amongst the general population in China means that caution should be taken in extrapolating these findings to the contemporary population in China. This is frustrating but when social developments are so rapid, even context-dependent information can be informative. Finally, this study included only mothers living with their children when interviewed, excluding, for example, those migrant working mothers. This may reduce the range of contrast in the study.

In conclusion, this study suggests that grandparental childcare is a factor for women's fertility that directly increases the likelihood of a second birth, but also delays the timing in the parity progression. Mothers' working status is negatively associated with the transition to a second birth and cancels out a considerable part of the influence of grandparental childcare on fertility. Thus, grandparents' involvement in childcare mainly supports Chinese working mothers. It does not provide a win-win solution to both increased fertility and female employment at the micro-level. However, the processes underlying interdependent life domains warrants further investigation.

## Acknowledgments

Link to code and reproducible materials: https://osf.io/vw9b3/?view_only=922bf2d5c2b74b5b9f5f0018d2d41add.

## Author Contributions

**Conceptualization:** Jing Zhang.

**Data curation:** Jing Zhang.

**Formal analysis:** Jing Zhang.

**Methodology:** Jing Zhang.

**Project administration:** Jing Zhang.

**Supervision:** Tom Emery.

**Visualization:** Jing Zhang.

**Writing – original draft:** Jing Zhang.

**Writing – review & editing:** Tom Emery.

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
