## [Decision Letter · Decision Letter 0]

13 Mar 2023

PONE-D-22-33080Grandparental Childcare and Second Births in ChinaPLOS ONE

Dear Dr. Emery,

Thank you for submitting your manuscript to PLOS ONE. After careful consideration, we feel that it has merit but does not fully meet PLOS ONE’s publication criteria as it currently stands. Therefore, we invite you to submit a revised version of the manuscript that addresses the points raised during the review process.

Two experts in the field have reviewed the manuscript noting in general its soundness and adequacy of methods. However, there are issues of interpretation of the findings listed by reviewer 1 that need to be addressed before it can be considered for publication.

We look forward to receiving your revised manuscript.

Kind regards,

José Antonio Ortega, Ph.D.

Academic Editor

PLOS ONE

Journal Requirements:

2. Please update your submission to use the PLOS LaTeX template. The template and more information on our requirements for LaTeX submissions can be found at http://journals.plos.org/plosone/s/latex

Reviewers' comments:

Reviewer's Responses to Questions

**Comments to the Author**

1. Is the manuscript technically sound, and do the data support the conclusions?

Reviewer #1: Partly

Reviewer #2: Yes

2. Has the statistical analysis been performed appropriately and rigorously? 

Reviewer #1: Yes

Reviewer #2: Yes

3. Have the authors made all data underlying the findings in their manuscript fully available?

Reviewer #1: Yes

Reviewer #2: Yes

4. Is the manuscript presented in an intelligible fashion and written in standard English?

Reviewer #1: Yes

Reviewer #2: Yes

5. Review Comments to the Author

Reviewer #1: *The points are presented in the order of appearance in the manuscript, not according to order of importance*

In the abstract and at a few points in the paper the authors write something to the effect of: “Those who use grandparental childcare are four times as likely to have a second child”….If the authors refer to 4 times higher ODDS of a second child, they should say so. Four times as likely seems to refer (incorrectly) to probability rather than odds. Also, when they say “30% slower transition to the second child”, it is not clear where that number comes from. In general, the results/findings are a bit difficult to interpret.

The authors write “Even though the one-child policy was in place until 2015, large parts of the population have enjoyed exemptions to this policy.”. This needs to be explained further, as it is key to understanding parity progression. Do exemptions vary by SES, place of residence (rural/urban), sex of first child, etc, and might these factors be related to grandparental care and women’s work? This could have important implications for the interpretations and conclusions based on the empirical findings.

In motivating Hypothesis 1, the authors neglect to discuss the potential endogeneity of grandparental childcare. Grandparental child care assistance may reflect difficulties that the parents experience in caring for their child, and thus may be indicative of more potential difficulty in having a second child. This should be mentioned, because, following this reasoning, it might be expected that grandparental childcare might be associated with LOWER odds of transition to the second birth, rather than higher odds, as per hypothesis 1.

The same point is relevant for the discussion of Hypothesis 2.

In the discussion of the motivation of Hypothesis 3, the authors neglect mentioning the idea that working mothers may have higher family income, which may ENCOURAGE the birth of a second child. This “income” effect may be particularly relevant given the high costs associated with children, and the point should be discussed because it implies a HIGHER odds of second birth among working mothers, rather than lower odds, as per hypothesis 3.

When discussing the methodology, the authors write that Cox models assume that the transition would “eventually” occur for all women, given a long enough follow-up. I think this is not a correct interpretation of the Cox models. Because the model takes data as censored beyond a certain age (e.g. 45), and there are no observations beyond duration of 16 years (as per the authors), the Cox model should reflect only patterns up to limited durations/ages. There is no "forever", and this makes no sense in the context of biological limitations on fertility.

More generally, if the authors want to make a methodological point regarding the limitation of Cox regression in comparison to the methods they employ (which they do in the conclusion), they should present results based on Cox models and compare them with the results from alternative models, to show the differences. Especially given the counter-intuitive results regarding grandparental care, in terms of both increasing odds of the transition, but also slowing down the speed of the transition, it would be more convincing if the authors showed that these results were robust to different model estimation strategies.

In the section on “Grandparental childcare, mother’s work and second childbirth”, the authors state that “Somewhat counterintuitively, a negative coefficient indicates that women are at higher risk of a second birth as the model estimates the likelihood of having a second child during the observation period.” This is very confusing because the upper panel of Table 2 is labelled as likelihood of NOT having the second child. Also the label on the lower panel of Table 2 is confusing because the label is “timing of failure”, which suggests that a positive coefficient should indicate faster progression, rather than slower progression as the authors state.

The authors write “More importantly, the value of the estimates in Model 3 increased in both positive and negative terms, particularly in the duration sub-model.” This sentence is unclear.

In the text, in connection with footnote 1, the authors write “Based on a comparison of the hazard ratios in different models, mother’s working status mediates 64% of the delay in the second birth in the group of mothers receiving grandparental childcare help”. It is unclear how this is possible because when you control for mother's work, the coefficient increases in absolute value, so how can mother’s work be mediating the effect of grandparental care?

In footnote 1, it is unclear why there is -1 in the denominator. Can the authors explain further?

The authors write “Care arrangements for children other than grandparents may be more amenable to compressed birth spacings than grandparental care.” It is unclear why this might be the case, and this requires further explanation.

The authors write “For example, the timing of a second child when grandparental care is involved

is now dependent on the preferences of three parties rather than two, increasing the potential

for delays.” This is not convincing as written and requires further development.

In discussion of Figure 3, the authors neglect mentioning that mother's working status may also affect grandparental childcare (in addition to the other way around)..... this should be addressed.

Also in Figure 3, why is the box labelled “NO second birth”? In the text, the authors write that they model that likelihood of having a second birth. Again, this is very confusing.

In the “Limitations” section, the authors write “This makes it possible that mothers’ working status

confounds part of the effect of grandparental childcare”. It is unclear what the authors mean by this in the current context.

Also in the “Limitations” section, the authors write “However, there could be selection bias and other unobserved heterogeneity”. This is not a helpful sentence. There is always that possibility. Authors need to explain that point more in the specific context at hand.

In the concluding section, the authors write “In conclusion, this study suggests that grandparental childcare is a factor for women’s fertility that directly increases the likelihood of a second birth, but also delays the timing in the parity progression. “ The point about delay does not appear consistent with the graphs based on the KM survival functions, in which grandparental childcare appears to be associated with faster parity progression. This point leads to some questioning regarding the robustness of the author’s findings to model specification. See point above regarding modeling strategies (e.g. Cox modeling).

With regard to Figure 1A, how can it be that more than 50% of women appear to have a second birth within 6 years, if the descriptive statistics (Table 1) indicate that it is only about 20-25%?

Reviewer #2: I agree with the authors that the topic of this research paper is important and deserves examination. The data used are appropriate and the data plan and analyses are also appropriate. The results largely support the hypotheses, with one exception, suggesting more research may be necessary on this topic. The manuscript is easy to read and demonstrates mastery of technical analysis and theoretical perspectives.

6. PLOS authors have the option to publish the peer review history of their article (what does this mean?). If published, this will include your full peer review and any attached files.

Reviewer #1: No

Reviewer #2: **Yes: **Toni Falbo

---

## [Author Response · Author response to Decision Letter 0]

19 Apr 2023

We would like to thank the reviewers for their comments. They have helped us to greatly clarify the contributions of the paper and we found their comments thoughtful and considered. We have included responses to reviewer one below.

Reviewer #1: *The points are presented in the order of appearance in the manuscript, not according to order of importance*

COMMENT 1: In the abstract and at a few points in the paper the authors write something to the effect of: “Those who use grandparental childcare are four times as likely to have a second child”….If the authors refer to 4 times higher ODDS of a second child, they should say so. Four times as likely seems to refer (incorrectly) to probability rather than odds. Also, when they say “30% slower transition to the second child”, it is not clear where that number comes from. In general, the results/findings are a bit difficult to interpret.

RESPONSE 1: We have corrected this in the abstract and the paper. We have repharsed the 30% figure as you are quite right that ‘slower’ does in fact mean 30% lower odds of transitioning in each period. The confusion over where this number comes from likely stems from the fact that you have to invert the odds to get the likelihood of having a child. So using the coefficient from model 4 to get: 1/e ^(0.35) = 0.7 or an odds ratio showing 30% lower odds for having a baby in each period relative to not. I hope this clarifies.

COMMENT 2: The authors write “Even though the one-child policy was in place until 2015, large parts of the population have enjoyed exemptions to this policy.”. This needs to be explained further, as it is key to understanding parity progression. Do exemptions vary by SES, place of residence (rural/urban), sex of first child, etc, and might these factors be related to grandparental care and women’s work? This could have important implications for the interpretations and conclusions based on the empirical findings.

RESPONSE 2: We have clarified that these exemptions were largely for parts of the rural population and non-han ethnic groups. We included the hukou as a key independent variable in the analysis due to this. We also included whether the mother was an only child. In terms of interpretation, it is indeed true that caution should be made when seeking to extrapolate these findings forward to post-reform period populations. To reflect this we have emphasised this in the limitations and stressed that without more contemporary and updated data, it is not yet possibly to fully extrapolate to contemporary populations.

COMMENT 3: In motivating Hypothesis 1, the authors neglect to discuss the potential endogeneity of grandparental childcare. Grandparental childcare assistance may reflect difficulties that the parents experience in caring for their child, and thus may be indicative of more potential difficulty in having a second child. This should be mentioned, because, following this reasoning, it might be expected that grandparental childcare might be associated with LOWER odds of transition to the second birth, rather than higher odds, as per hypothesis 1.

RESPONSE 3: We accept this and have added it as an alternative perspective in the discussion of the 1st hypotheses. However, this is more common in western contexts where grandparental childcare support is seen as a signifier of auxiliary support. In a Chinese cultural context, grandparental childcare is more of a default and expected when practical constraints allow. This is why we had omitted this counterhypothesis.

COMMENT 4: The same point is relevant for the discussion of Hypothesis 2.

RESPONSE 4: We have also added it there.

COMMENT 5: In the discussion of the motivation of Hypothesis 3, the authors neglect mentioning the idea that working mothers may have higher family income, which may ENCOURAGE the birth of a second child. This “income” effect may be particularly relevant given the high costs associated with children, and the point should be discussed because it implies a HIGHER odds of second birth among working mothers, rather than lower odds, as per hypothesis 3.

RESPONSE 5: We have added this consideration to the discussion of hypotheses 3. 

COMMENT 6: When discussing the methodology, the authors write that Cox models assume that the transition would “eventually” occur for all women, given a long enough follow-up. I think this is not a correct interpretation of the Cox models. Because the model takes data as censored beyond a certain age (e.g. 45), and there are no observations beyond duration of 16 years (as per the authors), the Cox model should reflect only patterns up to limited durations/ages. There is no "forever", and this makes no sense in the context of biological limitations on fertility.

RESPONSE 6: Whilst it is correct that the cox model uses censoring to address those who do not have a second child, this prevents assessment of the speed at which this transition comes about. This has been stipulated and demonstrated elsewhere: https://link.springer.com/article/10.1007/s10680-009-9201-2. 

COMMENT 7: More generally, if the authors want to make a methodological point regarding the limitation of Cox regression in comparison to the methods they employ (which they do in the conclusion), they should present results based on Cox models and compare them with the results from alternative models, to show the differences. Especially given the counter-intuitive results regarding grandparental care, in terms of both increasing odds of the transition, but also slowing down the speed of the transition, it would be more convincing if the authors showed that these results were robust to different model estimation strategies.

RESPONSE 7: We do not want to make this point. This point has already been shown. Using the cox model would not provide any clarification on the issue of transition speed. https://link.springer.com/article/10.1007/s10680-009-9201-2

COMMENT 8: In the section on “Grandparental childcare, mother’s work and second childbirth”, the authors state that “Somewhat counterintuitively, a negative coefficient indicates that women are at higher risk of a second birth as the model estimates the likelihood of having a second child during the observation period.” This is very confusing because the upper panel of Table 2 is labelled as likelihood of NOT having the second child. Also the label on the lower panel of Table 2 is confusing because the label is “timing of failure”, which suggests that a positive coefficient should indicate faster progression, rather than slower progression as the authors state.

RESPONSE 8: This is a typo in the text and has now been corrected to read: “Somewhat counterintuitively, a negative coefficient indicates that women are at higher risk of a second birth as the model estimates the likelihood of not having a second child during the observation period”

The interpretation of the second panel is correct however and a positive coefficient does indicate a slower transition to second birth. We have added a sentence to clarify this: “This is because a positive coefficient indicates an increased likelihood of not having a childhood in a specific time period, given that they do have a second child over the observation period.”

COMMENT 9: The authors write “More importantly, the value of the estimates in Model 3 increased in both positive and negative terms, particularly in the duration sub-model.” This sentence is unclear.

RESPONSE 9: We tried to clarify this statement by more clearly referring to the size of the coefficients: “The size of the coefficients in Model 3 increased in both positive and negative terms, particularly in the duration sub-model”

COMMENT 10: In the text, in connection with footnote 1, the authors write “Based on a comparison of the hazard ratios in different models, mother’s working status mediates 64% of the delay in the second birth in the group of mothers receiving grandparental childcare help”. It is unclear how this is possible because when you control for mother's work, the coefficient increases in absolute value, so how can mother’s work be mediating the effect of grandparental care? In footnote 1, it is unclear why there is -1 in the denominator. Can the authors explain further?

RESPONSE 10: We agree that the use of the term mediation is confusing here, especially given the presentation of figure 3. We have therefore reframed this as a ‘suppression’ effect and corrected the footnote accordingly. It now represents the degree to which the effect of grandparenting is suppressed by not controlling for female employment.

COMMENT 11: The authors write “Care arrangements for children other than grandparents may be more amenable to compressed birth spacings than grandparental care.” It is unclear why this might be the case, and this requires further explanation.

RESPONSE 11: Many thanks for highlighting this. We have tried to clarify this statement by referring explicitly to other forms of childcare such as private childcare that are not reliant on intergenerational solidarity and complex relationships dynamics.

COMMENT 12: The authors write “For example, the timing of a second child when grandparental care is involved

is now dependent on the preferences of three parties rather than two, increasing the potential

for delays.” This is not convincing as written and requires further development.

RESPONSE 12: We have elaborated and substantiated this point: For example, the timing of a second child when grandparental care is involved depends on the preferences of three parties rather than two, increasing the potential for delays. This would be particularly true if each party (mother, father, and grandparent) holds a veto on the decision of when to have another child. The timing must then be agreed upon by all three parties, and there is an increased likelihood of deferment. This could be particularly true if the timing decision now must also align with the life course developments and responsibilities of the grandparent.

COMMENT 13: In discussion of Figure 3, the authors neglect mentioning that mother's working status may also affect grandparental childcare (in addition to the other way around)..... this should be addressed.

RESPONSE 13: We agree with the substantive point but this was addressed in the manuscript directly below Figure 3. However, we didn’t not specifically refer to the possibility of maternal employment increasing grandparental care and we have now included this: “The design of this study cannot establish the direction of causality. One of the limitations of this study is that information on grandparental childcare and mother’s working status was collected at the same time point in CFPS. This makes it possible that mothers’ working status determines the level of grandparental childcare. Furthermore, both childcare and mothers’ labour force participation are processes that unfold over time rather than as static phenomena. This also means that it remains unclear whether grandparental childcare leads to maternal employment or vice versa. In reality, these are likely two interdependent and mutually reinforcing processes. The current circumstances are built on past experiences, and experiences in different life domains are influenced by each other, which is known as ‘interdependencies of parallel careers’ (Dykstra & van Wissen, 1999). Due to limited data, we cannot apply a counterfactual causal mediation analysis (Bijlsma & Wilson, 2020) with the split-population models presented in this paper. Rather than seeking a warrant for making absolute claims, the issue of causal interrelationships between these factors and the pathways of the direct and indirect influences of childcare and work on fertility over time is reserved for future work, where suitable data are more readily available.”

COMMENT 14: Also in Figure 3, why is the box labelled “NO second birth”? In the text, the authors write that they model that likelihood of having a second birth. Again, this is very confusing.

RESPONSE 14: As stated above, this was an error in the text that is now corrected, apologies.

COMMENT 15: In the “Limitations” section, the authors write “This makes it possible that mothers’ working status confounds part of the effect of grandparental childcare”. It is unclear what the authors mean by this in the current context.

RESPONSE 15: We have corrected and clarified this in the rewrite above.

COMMENT 16: Also in the “Limitations” section, the authors write “However, there could be selection bias and other unobserved heterogeneity”. This is not a helpful sentence. There is always that possibility. Authors need to explain that point more in the specific context at hand.

RESPONSE 16: We have specified what was meant here: “However, there could be selection bias and other unobserved heterogeneity in the types of grandparental childcare given. It is unlikely that the findings would be similar for a lower level of grandparental childcare involvement, or for using other measures such as co-residence, financial support from grandparents, contact or relationship quality between mother and grandparents.”

COMMENT 17: In the concluding section, the authors write “In conclusion, this study suggests that grandparental childcare is a factor for women’s fertility that directly increases the likelihood of a second birth, but also delays the timing in the parity progression. “ The point about delay does not appear consistent with the graphs based on the KM survival functions, in which grandparental childcare appears to be associated with faster parity progression. This point leads to some questioning regarding the robustness of the author’s findings to model specification. See point above regarding modeling strategies (e.g. Cox modeling).

RESPONSE 17: This is precisely the point of using the split population model. Using an approach that includes the whole population in the analysis clouds the discussion of timing and whether an event actually happens. We have added the following text to clarify this contribution:

“Using the split population model, the analysis shows that whilst grandparental care appears to accelerate transition to second birth in the population as a whole, it actually leads to a deference of transition when we look exclusively at the population that ultimately transitions. This indicates that grandparental care simultaneously increases the likelihood of transition to a second child and defers it”.

---

## [Editor Report · Decision Letter 1]

18 May 2023

Grandparental Childcare and Second Births in China

PONE-D-22-33080R1

Dear Dr. Emery,

We’re pleased to inform you that your manuscript has been judged scientifically suitable for publication and will be formally accepted for publication once it meets all outstanding technical requirements.

Kind regards,

José Antonio Ortega, Ph.D.

Academic Editor

PLOS ONE

Additional Editor Comments (optional): Among the 2 previous reviewers, reviewer 2 was recommending to accept. Reviewer 1, who posed several issues regarding, in particular, interpretation, was not available at this time, but the editor assesses that the issues raised have been addressed satisfactorily.
---

## [Editor Report · Acceptance letter]

26 May 2023

PONE-D-22-33080R1 

Grandparental Childcare and Second Births in China 

Dear Dr. Emery:

I'm pleased to inform you that your manuscript has been deemed suitable for publication in PLOS ONE. Congratulations! Your manuscript is now with our production department. 

Kind regards, 

on behalf of

Dr. José Antonio Ortega 

Academic Editor

PLOS ONE